# FALSE, MISLEADING, AND UNFOUNDED STATEMENTS IN A RECENT TPAMI PUBLICATION

## ABSTRACT

A recent TPAMI response raises issues with the contents of a recent TPAMI comment and the data collection underlying that comment. Several of the claims in that response are unfounded, inaccurate, misleading, false, invalid, or unsupported, as demonstrated by text in the comment and cited work, and new analyses that we report. The response further ignores key components of the work that it responds to.

## 1 INTRODUCTION

A recent response (Palazzo et al., 2024) raises issues with a recent comment (Bharadwaj et al., 2023) and the data collection (Ahmed et al., 2021) underlying that comment. Several of the claims in Palazzo et al. (2024) are unfounded, inaccurate, misleading, false, invalid, or unsupported, as demonstrated by text in Bharadwaj et al. (2023) and Ahmed et al. (2021), and new analyses that we report. Palazzo et al. (2024) further ignore key components of Bharadwaj et al. (2023) and Ahmed et al. (2021). We clarify these below.

## 2 SIGNAL BLEEDING ACROSS TRIALS

Palazzo et al. (2024) claim that the interleaved design used by Bharadwaj et al. (2023) and Ahmed et al. (2021) allows brain activity measured by EEG to bleed between adjacent trials.[1]

> *On the contrary, interleaved-design experiments introduce several confounds that may suppress the very response that one would hope to classify with machine learning methods. Indeed, object recognition in humans tends to last many hundreds of milliseconds (especially when the items change rapidly). This means that components such as the P300 and the N400 may still be processing the item from one class, when an item from the next class is presented [14]. This response overlap certainly results in the signal bleeding into the subsequent trial.*
>
> Palazzo et al. (2024)

While this may be true for designs such as those used by Spampinato et al. (2017), Kavasidis et al. (2017), and Palazzo et al. (2017; 2018; 2020a;b; 2021), Li et al. (2021), and Ahmed et al. (2022) where trials had duration 0.5 s and did not have any blanking between trials, the trials in Ahmed et al. (2021), one of the datasets used by Bharadwaj et al. (2023), had duration 2 s with 1 s blanking between trials.

> *Each run started with 10 s of blanking, followed by 400 stimulus presentations, each lasting 2 s, with 1 s of blanking between adjacent stimulus presentations, followed by 10 s of blanking at the end of the run.*
>
> Bharadwaj et al. (2023)

In the design of Ahmed et al. (2021), one of the datasets used by Bharadwaj et al. (2023), the items do not change rapidly and the 1 s blanking between trials is likely to preclude significant signal bleeding between adjacent trials. Thus the claim by Palazzo et al. (2024) that the interleaved design used by Bharadwaj et al. (2023) and Ahmed et al. (2021) "certainly results in the signal bleeding into the subsequent trial" is unfounded.

---

[1]All citation numbers in quoted text are those in the original.

## 3  SUBJECT ATTENTIVENESS

Palazzo et al. (2024) claim that block designs make the class more salient than interleaved designs and raised a concern about the attentiveness of the subject in Ahmed et al. (2021).

> *Additionally, when items are presented in a block, it is possible to make the class very salient (i.e., the participant will notice that they have viewed 50 dogs in a row), whereas the interleaved design obscures the point of the study. In this case, if the subjects were even mildly inattentive, they would certainly fail to think about the current class, something that is far harder to miss in the block-design. Obscuring the class like Bharadwaj et al. did, without requiring an overt response from the subject, calls into question if the subject was even paying attention to the stimuli, whereas an overt response forces the subject to attend to and more fully process the stimuli to the class level [14].*

Palazzo et al. (2024)

That may be an issue when presenting stimuli for 0.5 s with no blanking between stimuli, but is likely to be less of an issue when presenting stimuli for 2 s with 1 s blanking between stimuli. But beyond this, Ahmed et al. (2021) report strong evidence that the subject did attend to the stimuli.

> *To check whether the subject consistently viewed the images presented, online trial averaging of the EEG data was performed in every session to obtain evoked responses that are phase-locked to the onset of the images. Data from two occipital channels (C31 and C32) were bandpass filtered in the 1–40 Hz range and epochs of 800 ms duration were segmented out synchronously following the onset of each image. Epochs with peak-to-trough fluctuations exceeding 100 $\mu$V were discarded and the remaining epochs were averaged together to yield an 800 ms-long evoked response. A clear and robust N1-P2 onset response pattern was discernible in the evoked response traces obtained in each of the 100 runs, consistent with the subject viewing the images as instructed. Note that all online averaging procedures (e.g., filtering) were done to data in a separate buffer; the raw unprocessed data from 96 channels was saved for offline analysis.*

Ahmed et al. (2021)

Further evidence of subject attentiveness is that Ahmed et al. (2021) report statistically significant classification accuracy as high as 7.3% and Bharadwaj et al. (2023) report statistically significant classification accuracy as high as 17.6% on a task where chance performance is 2.5%. Given the randomized nature of the design, this would not be possible if the subject did not attend to the stimuli. Thus the concern raised by Palazzo et al. (2024) about the subject in Ahmed et al. (2021) as to whether "the subject was even paying attention to the stimuli" is unfounded.

## 4  SESSION LENGTH

Palazzo et al. (2024) claim that the data collection underlying Spampinato et al. (2017), Kavasidis et al. (2017), and Palazzo et al. (2017; 2018; 2020a;b; 2021) and had sessions lasting about 4 minutes.

> *In the data collection carried out by Bharadwaj et al. in [7] and also employed in [1], the authors state that a subject underwent stimuli exposition for over 20 minutes, instead of about 4 minutes in [3].*

Palazzo et al. (2024)

Similar claims are made six times in Palazzo et al. (2020b). However Spampinato et al. (2017, Table 1), Kavasidis et al. (2017, Table 1), and Palazzo et al. (2017, Table 1) state that session running time was 350 s, *i.e.*, 5 minutes and 50 s. This is more-or-less consistent with the protocol described in Spampinato et al. (2017), Kavasidis et al. (2017), and Palazzo et al. (2017) where each session contained 10 blocks, each block contained 50 stimuli, each stimulus lasted 0.5 s, and blocks were separated by 10 s blanking. Thus the claim in Palazzo et al. (2024) that the data collection in Spampinato et al. (2017) took "about 4 minutes" is inaccurate.

## 5 Cross-Subject Variability

Palazzo et al. (2024) claim that Li et al. (2021) observe large subject-to-subject variability in classification accuracy.

> *Even Bharadwaj et al. in [21] observe large subject-to-subject variability in their reported results, as classification performance of their own proposed method varies from 37.80% to 70.50% (Table 4 in [21], and Tables 21–25 in [21]'s appendix).*
>
> Palazzo et al. (2024)

Li et al. (2021, Tables 4, 21–25) discuss block runs. The central claim of Li et al. (2021) is that the block runs suffer from a temporal confound and thus one cannot draw any conclusions about stimulus processing from these block runs. In contrast, to assess cross-subject variability in Li et al. (2021), one needs to limit consideration to Li et al. (2021, Tables 5, 26–30) because these report randomized trials on image stimuli and the full 96 channels with bandpass filtering. These tables do not differ from chance in a statistically significant fashion. Thus the claim of Palazzo et al. (2024) that Li et al. (2021) "observe large subject-to-subject variability in their reported results" is misleading.

## 6 Single Subject

Palazzo et al. (2024) claim that the supertrial method of Bharadwaj et al. (2023) was applied to only a single subject.

> *A recent comments paper [1] by Bharadwaj et al. discusses the results presented in [2], claiming that the above-chance accuracy reported by that method is due to confounds in the experimental design (from [3]). In order to support that claim, Bharadwaj et al. propose a new dataset that is, according to them, free from those confounds. The key aspect of this dataset is that samples — or, as they call them, "supertrials", borrowing terminology from [4] — are obtained by averaging a set of trials collected during EEG recording for a single subject.*
>
> Palazzo et al. (2024)

They further state:

> *The dataset used by Bharadwaj et al., introduced in [7], is the result of EEG data collection on one subject only. Single-subject analysis is critical mainly because EEG data are known to be highly replicable within a person [14], but also highly specific from person to person [14], [20].*
>
> Palazzo et al. (2024)

This, however, ignores the fact that Bharadwaj et al. (2023) report not only the results of a supertrial analysis on the single-subject data from Ahmed et al. (2021), but also on the data from Li et al. (2021) on six subjects.

> *We repeat this same method to all six subjects of the image rapid event data from Li et al. [10] and replicate the study of Ahmed et al. [2, inline unnumbered table 9] with supertrials instead of trials, with five-fold leave-one-portion-out cross validation.*
>
> Bharadwaj et al. (2023)

These results are reported in the right half of Bharadwaj et al. (2023, Table 1). Bharadwaj et al. (2023) further state:

> *Here, we form supertrials by aggregating trials from a single subject. One could form supertrials by aggregating trials from multiple subjects.*
>
> Bharadwaj et al. (2023)

Bharadwaj et al. (2023) report results for a total of seven subjects: the left half of Bharadwaj et al. (2023, Table 1) reports results on one subject and the right half reports results on six subjects. Thus the claim of Palazzo et al. (2024) that "The dataset used by Bharadwaj et al., introduced in [7], is the result of EEG data collection on one subject only" is false.

## 7   EFFECT OF SUPERTRIALS ON SIGNAL SPECTRUM

Palazzo et al. (2024) claim that the supertrial method of Bharadwaj et al. (2023) attenuates higher-frequency bands in the signal:

> *Interestingly, EEGNet outperforms EEGChannelNet at lower frequency bands, while our approach performs better at higher frequency bands, thus confirming the findings of [2]. Thus, EEGChannelNet works better at higher frequencies. However, higher frequencies are unavoidably attenuated by the supertrial method, proposed by [1]. Averaging trials acts as a low pass filter (high frequencies rarely align temporally; therefore phase differences lead to averaging out over trials [14]). Simply put, the authors explicitly test the model using low frequency information, which we previously reported to reduce classification accuracy (as shown in [2], low frequency classification accuracy of EEGChannelNet is 30 percent lower w.r.t. high frequency classification). Supertrials necessarily result in the averaging out of information with inconsistent phase but significant power in a specific frequency band, which still contains useful neural information [14].*
>
> Palazzo et al. (2024)

and this penalizes EEGChannelNet.

> *Additionally, their specific supertrial setup seems designed to penalize EEGChannelNet [2], since it has been shown to exploit high-frequency information, which are practically suppressed by sample averaging.*
>
> Palazzo et al. (2024)

Bharadwaj et al. (2023) state:

> *Here, we aggregate supertrials by unweighted average in the time domain. One could average in the frequency domain, potentially considering only certain bands (e.g., induced responses), weighting some samples or bands more than others, or more generally averaging some nonlinear transform, learned or hard-coded, of single trials.*
>
> Bharadwaj et al. (2023)

Now, we repeat the analyses of Bharadwaj et al. (2023) on the data from Ahmed et al. (2021), constructing supertrials by averaging in the frequency domain. We do this by performing an FFT on each sample, averaging the magnitude and phase of the samples independently, and performing an inverse FFT on the average. This is done independently on each channel.

Fig. 1 plots the spectra for the raw trials and supertrials of various sizes $N$, averaged over (super)trial and channel. It can be seen that this does not attenuate higher-frequency components. In fact, it amplifies them.

We further repeat the analysis of Bharadwaj et al. (2023, Table 1 left) on the data from Ahmed et al. (2021) using this supertrial averaging method (Table 1). EEGChannelNet is still at chance, while SVM, 1D CNN, EEGNet, and SyncNet are still above chance for various size supertrials, validating the original claim of Bharadwaj et al. (2023). Thus the claim by Palazzo et al. (2024) that "Supertrials necessarily result in the averaging out of information with inconsistent phase but significant power in a specific frequency band, which still contains useful neural information [14]" is invalid.

Beyond this, Bharadwaj et al. (2023) did not develop the supertrial method; they simply employed methods of Isik et al. (2014), Cichy et al. (2016), Greene & Hansen (2020), and Zheng et al. (2020a).

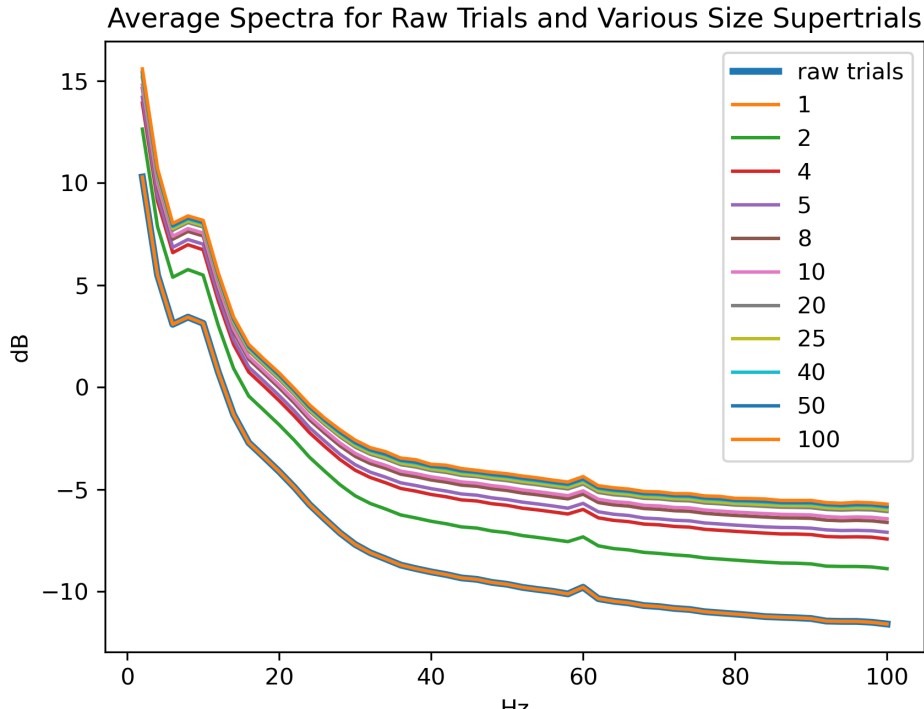

Figure 1: Spectra for the raw data from Ahmed et al. (2021) and various sizes of supertrials constructed by averaging in the frequency domain.

Table 1: Replication of the analysis from Bharadwaj et al. (2023, Table 1 left) for various sizes $N$ of supertrials. Starred values indicate statistical significance above chance ($p < 0.005$) by a binomial cmf. Note that when $N$ gets larger, the number of test samples gets smaller, increasing quantization noise in the accuracy estimates, thus requiring higher accuracy to achieve significance.

| $N$ | LSTM | $k$-NN | SVM | MLP | 1D CNN | EEGNet | SyncNet | EEGChannelNet |
|---|---|---|---|---|---|---|---|---|
| 1 | 2.2% | 2.1% | 5.5%* | 2.5% | 5.5%* | 7.1%* | 2.5% | 2.5% |
| 2 | 2.5% | 2.3% | 5.4%* | 2.4% | 5.0%* | 7.9%* | 2.7% | 2.5% |
| 4 | 2.4% | 2.5% | 6.3%* | 2.6% | 6.9%* | 8.7%* | 3.7%* | 2.5% |
| 5 | 2.1% | 2.4% | 6.0%* | 2.7% | 7.5%* | 7.0%* | 3.2%* | 2.4% |
| 8 | 2.3% | 2.4% | 3.2%* | 2.4% | 5.9%* | 9.5%* | 3.4%* | 2.4% |
| 10 | 2.2% | 2.1% | 2.6% | 2.4% | 4.5%* | 7.9%* | 3.2%* | 2.6% |
| 20 | 1.5% | 2.0% | 2.4% | 2.7% | 2.3% | 7.9%* | 3.0% | 2.9% |
| 25 | 3.4% | 2.1% | 2.3% | 2.3% | 2.9% | 3.5% | 2.6% | 2.6% |
| 40 | 2.2% | 2.7% | 2.2% | 2.3% | 2.0% | 2.6% | 3.4% | 1.7% |
| 50 | 2.1% | 3.0% | 2.8% | 2.5% | 2.8% | 3.1% | 3.6% | 2.4% |
| 100 | 4.0% | 1.5% | 3.5% | 3.3% | 3.0% | 5.3%* | 2.8% | 2.8% |

Since this work all predates Bharadwaj et al. (2023), and some of this work even predates Spampinato et al. (2017), Kavasidis et al. (2017), and Palazzo et al. (2017; 2018; 2020a;b; 2021), Bharadwaj et al. (2023) could not have designed the supertrial setup to penalize EEGChannelNet. Thus the claim by Palazzo et al. (2024) that "their specific supertrial setup seems designed to penalize EEGChannelNet [2]" is inaccurate.

.

## 8  CONFOUNDS

Palazzo et al. (2024) claim that interleaved-design experiments (aka randomized stimulus presentation order) introduce several confounds.

> *On the contrary, interleaved-design experiments introduce several confounds that may suppress the very response that one would hope to classify with machine learning methods.*
>
> Palazzo et al. (2024)

It is not clear what "several confounds" refers to. Nonetheless, none of the concerns raised by Palazzo et al. (2024) about Bharadwaj et al. (2023) and Ahmed et al. (2021) constitute confounds, even if they were true. According to APA (2024), a confound is:

> *in an experiment, an independent variable that is conceptually distinct but empirically inseparable from one or more other independent variables. Confounding makes it impossible to differentiate that variable's effects in isolation from its effects in conjunction with other variables.*
>
> APA (2024)

Palazzo et al. (2024) misuse the term "confound".

The protocol of Spampinato et al. (2017), Kavasidis et al. (2017), Palazzo et al. (2017; 2018; 2020a;b; 2021), and the block runs of Li et al. (2021) and Ahmed et al. (2022), does suffer from a confound, namely, a correlation between stimulus class and time since the start of the run, essentially a clock embedded in the signal. As a result, it is impossible to determine whether the classifier is classifying stimulus class or the embedded clock. This temporal confound excessively *overestimates* the classification accuracy. Even if they were true, the concerns raised by Palazzo et al. (2024) about Bharadwaj et al. (2023) and Ahmed et al. (2021) only would reduce the quality of the data and *underestimate* the classification accuracy. Any potential limitations of the interleaved-design experiments would not constitute "confounds." Thus the claim by Palazzo et al. (2024) that "interleaved-design experiments introduce several confounds" is false.

Palazzo et al. (2024) claim that the protocol of Spampinato et al. (2017), Kavasidis et al. (2017), and Palazzo et al. (2017; 2018; 2020a;b; 2021) does not suffer from a confound.

> *The claim that classification in block-design experiments mainly relies on temporal correlations has already been addressed in [13], where we showed that:*
> - *Models are not able to classify samples from a rapid-design setup when block-level labels are artificially assigned;*
> - *Samples collected during blank screens between two blocks are unlikely to be classified as coming from the class before or after the blank screen.*
>
> Palazzo et al. (2024)

This line of reasoning exhibits a logical fallacy. According to Frost (2024):

> *You can't prove a negative! [. . . ] If your test fails to detect an effect, it's not proof that the effect doesn't exist. It just means your sample contained an insufficient amount of evidence to conclude that it exists.*
>
> Frost (2024)

The presence of a confound in the protocol used by Spampinato et al. (2017), Kavasidis et al. (2017), and Palazzo et al. (2017; 2018; 2020a;b; 2021) is clearly demonstrated by the incorrect block-level labels experiment reported in Li et al. (2021, Tables 9 and 10) wherein it is shown that classifiers can decode incorrect block-level class labels that are unrelated to the actual stimuli used to elicit EEG response from trials with randomized stimulus presentation order.

Luck (2014) references twenty three discussions of confounds in the index. Among them, Luck (2014, p. 133) states:

*Ignorance and Lack of Imagination When someone says, "I can't imagine how that little confound could explain my results," this is a case of a general logical fallacy that philosophers call the argument from ignorance. In fact, it's a special case that is called (with a touch of humor) the argument from lack of imagination. The fact that someone can't imagine how a confound could produce a particular effect might just mean that the person doesn't have a very good imagination! I myself have occasionally used the "I can't imagine how . . . " type of reasoning and then found that I was suffering from a lack of imagination (see, e.g., box 4.5). But now that I realize that this is not a compelling form of argument, I usually catch myself before I say it.*

Luck (2014)

Palazzo et al. (2020b) (reference [13] in Palazzo et al. (2024)) offers two analyses in attempt to support their claim of a lack of a temporal confound in the data of Spampinato et al. (2017), Kavasidis et al. (2017), and Palazzo et al. (2017; 2018; 2020a;b; 2021). Palazzo et al. (2020b, Table 2) report an analysis whereby models are trained on BDVE, the original data used by Spampinato et al. (2017), Kavasidis et al. (2017), and Palazzo et al. (2017; 2018; 2020a;b; 2021), and tested on BDB, a dataset constructed from EEG collected when subjects viewed blank screens.

*The neural signals recorded between each pair of classes,* i.e., *the **BDB dataset**, can help address this question. Since the neural data in response to the blank screen is equidistant in time from two classes, a strong temporal correlation would result in significantly greater than chance classification of that data as either the class before or the class after the blank screen. Thus, we verify whether a model trained on the block-design **BDVE** dataset would classify blank screen segments as either the preceding or subsequent class. Finding near chance level classification accuracy here would indicate little to no impact of a temporal correlation. To assess the temporal correlation we assign two class labels to each blank segment in the BDB dataset, corresponding to the preceding class and the following class. Then, for each of the models trained on the BDVE dataset and whose results are given in Table 1, we compute the classification accuracy of the BDB dataset as the ratio of blank segments classified as either one of the corresponding classes. Results are shown in Table 2, and reveal that all methods are at or slightly above chance accuracy (i.e., 5%, since for each segment has two possible correct options out of the 40 classes).* **This seems to be a clear indication that temporal correlation in [2]'s data is minimal, suggesting that block design experiments (when properly pre-processed) are suitable for classification studies.**

Palazzo et al. (2020b)
(Emphasis in the original highlighted in bold.)

First note that Palazzo et al. (2020b, Table 2) do indeed report finding a temporal confound in the data of Spampinato et al. (2017), Kavasidis et al. (2017), and Palazzo et al. (2017; 2018; 2020a;b; 2021). Second, this analysis does not accurately assess the temporal confound in the original results in Spampinato et al. (2017), Kavasidis et al. (2017), and Palazzo et al. (2017; 2018; 2020a;b; 2021), as described below.

Li et al. (2021) discuss two kinds of temporal confound, one where the training and test sets come from the same blocks of the same runs (Li et al., 2021, Table 6) and one where the training and test sets comes from temporally correlated blocks of two different runs (Li et al., 2021, § 3.7, Table 15). Note that the former has considerably higher accuracy than the latter, yet both are considerably above chance. This suggests that there is a strong temporal correlation within the blocks of the same run and a weaker, but still present, temporal correlation between temporally correlated blocks of different runs.

The BDB analysis of Palazzo et al. (2020b) measures the latter, not the former. It is thus expected that the temporal correlation will be less than that present in Spampinato et al. (2017), Kavasidis et al. (2017), and Palazzo et al. (2017; 2018; 2020a;b; 2021) which is of the former kind. Thus, the claims in Palazzo et al. (2020b), that the "temporal correlation in [2]'s data is minimal" and "that block design experiments (when properly pre-processed) are suitable for classification studies", and the claim in Palazzo et al. (2024), that "The claim that classification in block-design experiments mainly relies on temporal correlations has already been addressed in [13]", are unfounded.

Further, the training and test samples in Spampinato et al. (2017) and Palazzo et al. (2017; 2018; 2020a;b; 2021) which come from the same block of the same run, have a uniformly distributed temporal distance between 0.5 s and 25 s whereas the test samples in BDB come from the blanking periods, not the stimulus periods. The temporal distance between the blanking periods and the corresponding stimulus periods varies uniformly between 25 s and 35 s. Palazzo et al. (2020b) state:

> *The data from these blank screens are particularly significant because, as claimed in [1], any contribution of a temporal correlation to classification accuracy should persist throughout the blank screen interval (i.e., the blank interval should be consistently classified above chance as either the class before or after the blank screen)*

Palazzo et al. (2020b)

Li et al. (2021) never claim this and we have no reason to believe that this is the case. It is likely that the temporal confound proceeds like a clock throughout the recording session. Palazzo et al. (2020b; 2024) misunderstand the nature of the confound in Spampinato et al. (2017), Kavasidis et al. (2017), and Palazzo et al. (2017; 2018; 2020a;b; 2021) reported by Li et al. (2021), Ahmed et al. (2021; 2022), and Bharadwaj et al. (2023). Thus the claim by Palazzo et al. (2024) that "any contribution of a temporal correlation to classification accuracy should persist throughout the blank screen interval (i.e., the blank interval should be consistently classified above chance as either the class before or after the blank screen)" is also not supported by the data.

Palazzo et al. (2020b, Table 4) report a second analysis, that replicates the analysis in Li et al. (2021, Table 9), whereby models are trained on BDVE, and tested on RDVE, a dataset collected with randomized trials (with half the samples per class than the datasets in either Li et al. 2021 or Spampinato et al. 2017, Kavasidis et al. 2017, and Palazzo et al. 2017; 2018; 2020a;b; 2021), but where the actual class labels are replaced with incorrect block-level labels. First note that Palazzo et al. (2020b, Table 4) do indeed report finding a temporal confound in the data of Spampinato et al. (2017), Kavasidis et al. (2017), and Palazzo et al. (2017; 2018; 2020a;b; 2021).

> *The classification accuracy, when using rapid-design data with incorrect block-level labels, is at most 9 percent points above chance, suggesting that the rapid design carries some small temporal correlations.*

Palazzo et al. (2020b)

Many factors could contribute to observing a smaller effect than that observed by Li et al. (2021), among them the fact that RDVE has half the samples per class. Thus the statement "at most 9 percent points above chance" is misleading when used to validate the use of data from Spampinato et al. (2017) and the results from Spampinato et al. (2017), Kavasidis et al. (2017), and Palazzo et al. (2017; 2018; 2020a;b; 2021).

Finally, Palazzo et al. (2024) state:

> *In [13], we further elucidate that the single-subject analysis is problematic, by demonstrating that pooling data across subjects accounts for inter-subject variability by reducing the subject-specific representation on the classifier. We show that the per-subject variability (measured in terms of standard deviation) decreases significantly when a classifier is trained using multiple subjects' data. Furthermore, this allows the model to focus on inter-subject discriminative features, reducing the bias due to possible temporal correlations that may exist in a single subject's neural responses. Thus, the large inter-subject differences must be overcome for any viable classification method. Importantly, averaged event-related data from a random sample of about 10 subjects tends to look highly similar to another random sample of 10 subjects [22], [14]. Failure to pool data across subjects would, again, only serve to increase the impact of any temporal correlation.*

Palazzo et al. (2024)

We have no reason to believe that the temporal correlation proceeds at the same rate in different subjects. Li et al. (2021, Table 8) assess this via a leave-one-subject-out analysis on the data from Spampinato et al. (2017), Kavasidis et al. (2017), and Palazzo et al. (2017; 2018; 2020a;b; 2021). The precipitous drop in classification accuracy from that reported by Spampinato et al.

(2017) and Palazzo et al. (2017; 2018; 2020a;b; 2021), while still "pooling training data across subjects," strongly suggests that the high accuracy reported by Spampinato et al. (2017) and Palazzo et al. (2017; 2018; 2020a;b; 2021) results from within-subject within-run temporal correlations that are absent across subjects. Thus the claim in Palazzo et al. (2024) "that pooling data across subjects accounts for inter-subject variability by reducing the subject-specific representation on the classifier" is unfounded.

We know of no successful results on performing cross-subject classification of EEG recordings from stimuli similar to those used in Spampinato et al. (2017), Kavasidis et al. (2017), and Palazzo et al. (2017; 2018; 2020a;b; 2021) that do not suffer from confounds. EEG data collection is resource limited. One can spend that resource collecting a smaller amount of data from multiple subjects or a larger amount of data from a single subject. Ahmed et al. (2021) decided to do the latter as cross-subject classification is infeasible at the current time and the intent was to assess the bounds of classification accuracy with a feasible data collection effort. The data collection from Ahmed et al. (2021) and Bharadwaj et al. (2023) was the largest known nonconfounded EEG dataset from stimuli similar to those used in Spampinato et al. (2017), Kavasidis et al. (2017), and Palazzo et al. (2017; 2018; 2020a;b; 2021) at the time of publication. Moreover, the classification accuracies were the highest known for nonconfounded data of that type at the time of publication. To our knowledge, both of these are still the case.

## 9 CONCLUSION

The key claims in Bharadwaj et al. (2023) are stated in the conclusion.

> *Palazzo et al. [14] claim that the data collected in Li et al. [10] lacks class information due to lack of subject attentiveness during long sessions, and that classification failure is based on this. [. . . ] Table I demonstrates that the data of Ahmed et al. [1] and Li et al. [10] do contain class information; it is just that some classifiers successfully extract it and some do not. Thus our results here refute their claim. Table I further demonstrates that:*
> - *With and without supertrials, EEGChannelNet yields chance accuracy on a nonconfounded dataset $20\times$ larger than that of [15].*
> - *For some amounts of supertrial aggregation, EEGNet and SyncNet yield above chance accuracy.*
>
> *This refutes the claim in [15] that EEGChannelNet outperforms EEGNet and SyncNet. Moreover, to the best of our knowledge, the classification accuracy of 17.5% obtained by EEGNet with $N = 20$ is the highest reported for a 40-class EEG classification task on ImageNet stimuli. Finally, this demonstrates that the datasets of Ahmed et al. [1] and Li et al. [10] do contain class information in the EEG signal; EEGNet, to some extent, and SyncNet, to a lessor extent, can extract that class information. EEGChannelNet cannot.*
>
> Bharadwaj et al. (2023)

Nothing in Palazzo et al. (2024) refutes that claim.

AUTHOR CONTRIBUTIONS

Removed for blind review.

ACKNOWLEDGMENTS

Removed for blind review.

ETHICS STATEMENT

This work debunks nearly one hundred published papers whose results are based on the same confound: a correlation between stimulus class and temporal drift. This confound has been found in eighteen available EEG datasets. Just as with an inconsistent set of axioms one can prove anything, a confounded dataset can be used to support any claim, even ones that are false or absurd. That is what many recent publications based on this confound do: things like generating high fidelity renderings of images, or even 3D CAD models of objects, from EEG recordings.

A research community, knowingly or unknowingly, has discovered that one can use confounded datasets to churn out a plethora of flawed results without reviewers noticing. They have also discovered that one can collect new confounded datasets to churn out even more flawed results without reviewers noticing. The temptation to do this is so strong that the community continues to do so four years after details of the confound were published.

It is conceivable that the flaws in these datasets may be a driving factor behind their frequent reuse. When a dataset is severely confounded, it becomes relatively easy to achieve an extremely high accuracy, which can in turn be used to support sensational claims, and ultimately directs further attention to the dataset. In business, this phenomenon is referred to as "the bad money drives out the good money."

More prominent exposure of these flawed methods and consequent false results will allow resources wasted on continued use of these confounded datasets and flawed methods to be reallocated. The debunked work also causes direct ongoing harm:

- grant proposals can be rejected due to preliminary results not being competitive with results demonstrating falsely-inflated performance based on confounded data or faulty methods;
- manuscripts can be rejected for the same reason;
- grants can be awarded based on false pretenses
- manuscripts can be accepted for the same reason;
- degrees can be awarded for the same reason;
- resources can be wasted attempting to replicate the debunked results;
- resources can be wasted having people read and review flawed papers, and learn flawed methods; and
- because the debunked work relates to brain-computer interfaces—whose primary application is helping people with disabilities (*e.g.*, paralysis) interact with the world—the harm caused is not merely scientific but also medical, with disproportionate impact on people with disabilities.

This work is significant for the following reasons:

- Nearly one hundred papers (An & Cho, 2016; Spampinato et al., 2016; Ben Said et al., 2017; Bozal Chaves, 2017; Kavasidis et al., 2017; Palazzo et al., 2017; Parekh et al., 2017; Spampinato et al., 2017; Zhang et al., 2017; Du et al., 2018; Fares et al., 2018; Kumar et al., 2018; Palazzo et al., 2018; Piplani et al., 2018; Tirupattur et al., 2018; Wang et al., 2018; Zhang & Liu, 2018; Zhang et al., 2018; Zhong et al., 2018; Du et al., 2019; Hwang et al., 2019; Jiang et al., 2019; Jiao et al., 2019; Long et al., 2019; Mukherjee et al., 2019; Uys, 2019; Wang et al., 2019; Cudlenco et al., 2020; Fares et al., 2020; Li et al., 2020; Palazzo et al., 2020a;b; Wang et al., 2020; Zheng et al., 2020b;c; Palazzo et al., 2021; Zheng & Chen, 2021; Ma et al., 2021; Mo et al., 2021; Jiang et al., 2021; Lee et al., 2021; Cavazza et al., 2022; Khaleghi et al., 2022; Lee et al., 2022; Mishra et al., 2022; Mishra, 2022; Scharnagl & Groth, 2022; Shimizu & Srinivasan, 2022; Ahmadieh et al., 2023; Bai et al., 2023; Du et al., 2023; Duan et al., 2023; Hasan & A, 2023; Imani et al., 2023; Lan et al., 2023; Lee et al., 2023; Liu et al., 2023; Singh et al., 2023; Song et al., 2023; Wahengbam et al., 2023; Zeng et al., 2023b;a; Fan et al., 2024; Ferrante et al., 2024a;b; Gou et al., 2024; Lei et al., 2024; Liu et al., 2024a;b; Luvsansambuu et al., 2024; Mishra et al., 2024; Mwata-Velu et al., 2024; Ngo et al., 2024; Palazzo et al., 2024; Pan et al., 2024; Qian et al., 2024; Singh et al., 2024; Tang et al., 2024; de la Torre-Ortiz et al., 2024; Yang & Liu, 2024; Ye et al., 2024; Zheng et al., 2024b;a; Zhu et al., 2024; Deng et al., 2025; Fares, 2025; Fu et al., 2025; Lopez et al., 2025; Mehmood et al., 2025; Singh et al., 2025; Xiang et al., 2025) draw flawed conclusions based on the confounded dataset from Spampinato et al. (2017) and datasets suffering from the same confound.
- A number of new datasets have been collected with this same confounded protocol (Gou et al., 2024; Pan et al., 2024; Zhu et al., 2024; Qian et al., 2024; Uys, 2019; Shimizu & Srinivasan, 2022; Liu et al., 2024b; Wang et al., 2019; 2020; Ma et al., 2021; Cudlenco et al., 2020; Zheng et al., 2024b; Cavazza et al., 2022; Luvsansambuu et al., 2024; Liu et al., 2023; Bai et al., 2023; Parekh et al., 2017).
- A number of these have been publicly released and are used by others. For example, Singh et al. (2023), Singh et al. (2024), and Lopez et al. (2025) use the dataset reported in Kumar

et al. (2018) and Duan et al. (2023), Singh et al. (2024), and Lopez et al. (2025) use the dataset reported in Ma et al. (2021).

- This is further egregious because Palazzo et al. (2020b; 2024) continue to claim that their dataset (Spampinato et al., 2017), and their results that were obtained with that dataset (Spampinato et al., 2017; Kavasidis et al., 2017; Palazzo et al., 2017; 2018; 2020a;b; 2021; 2024), are valid, despite the refutations in Li et al. (2021), Ahmed et al. (2021; 2022), and Bharadwaj et al. (2023), in part, because of the arguments in Palazzo et al. (2024).

- This has been used to justify continued publication of a large and growing body of flawed work based on confounded datasets (Cavazza et al., 2022; Khaleghi et al., 2022; Lee et al., 2022; Mishra et al., 2022; Mishra, 2022; Scharnagl & Groth, 2022; Shimizu & Srinivasan, 2022; Ahmadieh et al., 2023; Bai et al., 2023; Du et al., 2023; Duan et al., 2023; Hasan & A, 2023; Imani et al., 2023; Lan et al., 2023; Lee et al., 2023; Liu et al., 2023; Singh et al., 2023; Song et al., 2023; Wahengbam et al., 2023; Zeng et al., 2023b;a; Fan et al., 2024; Ferrante et al., 2024a;b; Gou et al., 2024; Lei et al., 2024; Liu et al., 2024a;b; Luvsansambuu et al., 2024; Mishra et al., 2024; Mwata-Velu et al., 2024; Ngo et al., 2024; Palazzo et al., 2024; Pan et al., 2024; Qian et al., 2024; Singh et al., 2024; Tang et al., 2024; de la Torre-Ortiz et al., 2024; Yang & Liu, 2024; Ye et al., 2024; Zheng et al., 2024b;a; Zhu et al., 2024; Deng et al., 2025; Fares, 2025; Fu et al., 2025; Lopez et al., 2025; Mehmood et al., 2025; Singh et al., 2025; Xiang et al., 2025) even after the confound became known through the work of Li et al. (2021), Ahmed et al. (2021; 2022), and Bharadwaj et al. (2023).

Current machine-learning conferences, and more generally, computer-science conferences and journals, are loathe to publish refutations. Observing this, Schaeffer et al. (2025) proposed that the field of machine-learning establish a "refutations and critiques" track in prominent conferences. While we applaud and support this proposal, the current lack of such a track should not be an impediment to publishing refutations. Scientific journals in other fields have long done so, often resulting in retraction of flawed work. Schaeffer et al. (2025) offer five example pieces of claimed flawed work in machine learning. Each is an individual paper. These pale in comparison to the flaws we uncover here: a systemic flaw of the entire peer review process across an entire field of inquiry, namely classification of stimulus image class from EEG recordings, that affects seventeen datasets and ninety one papers. Moreover, none of the five examples in Schaeffer et al. (2025) are egregious; here the authors of the flawed work continue to argue for its validity despite four refereed refutations and fifty new flawed papers have been published subsequent to these four refereed refutations. This argues for the need to make the community aware of the severity of the issue.

REPRODUCIBILITY STATEMENT

The raw data that produced these results is available at `https://dx.doi.org/10.21227/bc7e-6j47`. Our code, which will be released upon publication, is built on top of the code in `https://dx.doi.org/10.21227/bc7e-6j47`.

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
