# OpenReview forum: "False, misleading, and unfounded statements in a recent TPAMI publication"
_ICLR.cc/2026/Conference — Submitted to ICLR 2026_

### Official Review · Reviewer_NVHk · 2025-10-17

**Soundness:** 1
**Presentation:** 1
**Contribution:** 1
**Rating:** 0
**Confidence:** 5

**Summary:**

I think this paper should have been desk rejected.
It is essentially an attack on another article published in TPAMI, without any context or scientific contribution. The topic (brain–computer interface) is also outside my area of expertise.

**Strengths:**

N/A

**Weaknesses:**

N/A

**Questions:**

N/A

**Details Of Ethics Concerns:**

This paper should have been desk rejected

It is essentially an attack on another article published in TPAMI, without any context or scientific contribution.

---

> ### Author Response · Authors · 2025-11-21
>
> Viewing this as "an attack on another article" misses the point. It is
> legitimate scientific debate. Science advances when flaws in prior
> work are pointed out in a sufficiently prominent venue for the
> community to become aware of the flaws. This opens the field up to
> correcting those flaws.

---

> > ### Comment · Reviewer_NVHk · 2025-11-26
> > **ack of authors' response**
> >
> > hi, i acknowledge authors' responses but it doe snot change my feedback on the paper

---

### Official Review · Reviewer_AAD2 · 2025-10-31

**Soundness:** 2
**Presentation:** 1
**Contribution:** 2
**Rating:** 0
**Confidence:** 5

**Summary:**

In 2021, the following paper was published in TPAMI: Palazzo et al., "Decoding Brain Representations by Multimodal Learning of Neural Activity and Visual Features," TPAMI, 43(11): 3833–3849 (2021). Since its publication, several papers have identified substantial flaws in this work. In particular, two comments papers appeared in TPAMI (Ahmed et al., 2022; Bharadwaj et al., 2023). In 2024, Palazzo et al. published a rebuttal in TPAMI presenting their response to these criticisms. The present submission is a continuation of this discourse, aiming to highlight the issues in Palazzo et al. (2024). Each of Sections 2–8 examines individual claims from Palazzo et al. (2024) and argues why they are invalid, partially providing support through experimental results.

**Strengths:**

The paper provides a critical discussion of several technical details of a previous TPAMI publication, following up on the subsequent comments and rebuttal papers that also appeared in TPAMI. Thus, it may help clarify the technical issues that have been raised.

It also contributes to increasing general awareness of these and similar methodological challenges in neuroimaging and other fields.

**Weaknesses:**

This is a very unusual submission. A substantial portion of the text largely reiterates claims from the Palazzo et al. (2024) paper, as well as from the subsequent comment and rebuttal papers. The arguments presented for each cited claim are very brief and lack sufficient detail and discussion. In its current form, the paper is not self-contained, which makes it difficult to properly assess its content.

It is highly questionable whether ICLR or similar conferences would be an appropriate venue for this submission, as it does not present any novel technical contributions. Considering the follow-up history of the original Palazzo et al. (2021) paper, TPAMI appears to be the more suitable venue for continuing this discussion.

**Questions:**

Why not consider contributing this text to TPAMI?

Alternatively, why not consider a suitable online forum established for this ongoing discussion, potentially moderated by an authoritative figure in the field? This would allow many researchers to participate, fostering a collaborative effort aimed at arriving at the truth. In this context, such a platform might be even much more effective than publishing the work as a standalone paper.

---

> ### Author Response · Authors · 2025-11-21
>
> Re "very brief and lack sufficient detail": Most of the claims made by
> Palazzo et al. (2024) are demonstrated to be false simply by citing
> appropriate passages of Li et al. (2021), Ahmed et al. (2021, 2022),
> and Bharadwaj et al. (2023).
>
> Re "not self-contained": That is why we included both the passages in
> Palazzo et al. (2024) and the refuting passages in Li et al. (2021),
> Ahmed et al. (2021, 2022), and Bharadwaj et al. (2023). No technical
> judgment is needed for most issues. They are in plain English. A few
> need supporting evidence. Those are provided in Fig. 1 and Table 1. §7
> presents the details of how these analyses were performed and how they
> refute specific passages in Palazzo et al. (2024). The arguments are
> straightforward.
>
> Re "suitable online forum": Schaeffer et al. (2025) argues against
> this and argues that ML venues should establish a refutations and
> critiques track. We discuss this briefly in lines 559–570.

---

> > ### Comment · Reviewer_AAD2 · 2025-11-25
> >
> > I completely agree that there should be a forum for these kinds of discussions, which are important for keeping our field in a healthy state. However, for this particular venue, I believe the program chairs should first make a meta-level decision about whether such papers conform to the submission policy. If they do, these papers should be placed in a dedicated special track with a different reviewing procedure.

---

### Official Review · Reviewer_R1sm · 2025-11-01

**Soundness:** 3
**Presentation:** 3
**Contribution:** 3
**Rating:** 0
**Confidence:** 3

**Summary:**

The paper addresses statements in a recent publication.

**Strengths:**

The paper seems to address a major shortcoming in the published literature.

**Weaknesses:**

As far as I can see, no research contribution is made and the paper is off topic for ICLR. It is also over the page limit. The paper should be sent to TPAMI.

**Questions:**

n/a

---

> ### Author Response · Authors · 2025-11-21
>
> How is this manuscript over the page limit? It is 9 pages. The material
> starting on line 472 is not counted toward the page limit.

---

> > ### Comment · Reviewer_R1sm · 2025-11-22
> >
> > The ethics statement does not address any potential ethics issues with the paper itself, but contains material that could (and should) be in the main paper. It is also substantially more than a page.

---

> > > ### Author Response · Authors · 2025-11-22
> > >
> > > https://iclr.cc/Conferences/2026/AuthorGuide
> > > does not limit the ethics statement to "potential ethics
> > > issues with the paper itself". It states
> > >
> > >    "Topics include, but are not limited to, studies that involve human
> > >    subjects, practices to data set releases, potentially harmful
> > >    insights, methodologies and applications, potential conflicts of
> > >    interest and sponsorship, discrimination/bias/fairness concerns,
> > >    privacy and security issues, legal compliance, and research integrity
> > >    issues (e.g., IRB, documentation, research ethics)."
> > >
> > > While it is true that it is more than a page, the author guide also
> > > states that it is optional and does not count toward the page
> > > limit. The author guide further allows an unlimited amount of
> > > supplementary material. One could reasonably consider this to be
> > > supplementary material. We would be happy to upload a new version
> > > moving the current ethics statement to after the references and
> > > relabeling it as supplementary material.
> > >
> > > The severity of the issue and the eggregious nature of the myriad
> > > offending publications suggests that this minor technicality not be an
> > > impediment to publication.

---

### Author Response · Authors · 2025-11-21
**Historical Background and Significance**

To understand this work's significance, consider this brief historical
overview.

Spampinato et al. (2017) introduced a block-designed dataset
("Perceive") and methods that claim to achieve extremely high accuracy
decoding stimulus image class from EEG recordings. This was amplified
by follow on papers (Kavasidis et al. 2017, Palazzo et al. 2018,
2020a, 2020b, 2021), many of which claim to do things like reconstruct
stimulus images from EEG recordings. Further, Tirupattur (2018) does
this with a fresh dataset (Kumar 2018) that has the same block design.

Li et al. (2021) debunked all of the above, demonstrating that the
Perceive dataset suffers from a block confound. EEG exhibits drift,
encoding a clock in the signal. Since Perceive was collected with all
and only stimuli of the same class being temporally adjacent, the
classifier can mistakenly classify the clock/drift instead of the
stimulus-related EEG response. Follow on papers (Ahmed et al. 2021,
2022, Bharadwaj et al. 2023) added novel independent confirmation of
the results of Li et al. (2021).

Despite this, Palazzo et al. (2020b, 2021, 2024) continue to argue
that their dataset is valid. At this point, there are over one hundred
papers that use the Perceive dataset, the Kumar (2018) dataset, or
other datasets that suffer from the same block confound. Many new
datasets have been collected with this same block confound, some of
which are becoming widely used. The vast majority of these were
published after the confound became known (Li et al. 2021). Some of
these are unaware of the confound. Others are aware, but dismiss it,
often based on the argument of Palazzo et al. (2020b, 2021, 2024).

That argument is what this manuscript refutes.

This confound has been extensively debated on blogs like reddit, yet
that too has not stopped the extensive publication of flawed work.

There are three distinct levels of severity of this issue, which
progressively support greater need for continued publication:

 1. Many authors are unaware of the confound, despite the fact that it
    was published in prominent venues (e.g., TPAMI, CVPR, NeurIPS) and
    continue to publish flawed work

 2. While many authors are aware of the confound, they nonetheless
    ignore the warning and continue to publish flawed work.

 3. Some authors dismiss the confound and actively argue for the
    community to continue to employ flawed methods.

---

### Author Response · Authors · 2025-11-21
**Re: Public debate**

Several reviewers commented that public debate of this issue is
inappropriate. We realize that this may be unconventional and uncommon
in the ML community. But it is common in most other scientific fields
(e.g., Brain and Behavioral Science, Psycoloquy, ...). Public debate
through publication is the well-established method for arriving at
scientific truth. Schaeffer (2025) have argued that a mechanism for
publishing critiques and refutations is sorely lacking in ML.

The vast majority of the reviews focus on the fact that they are
unconventional. Essentially none of
the reviews discuss any technical flaws in these submissions. We
would be happy to discuss and address any technical flaws.

---

> ### Comment · Reviewer_R1sm · 2025-11-22
>
> Thank you for your reply. I'm certainly not saying that public debate of this issue is inappropriate, I'm saying that this doesn't seem to fall within any of the categories that ICLR solicits submissions for. Further, at least I'm not familiar enough with the offending paper to judge the claims within the short amount of time allocated for ICLR reviews.

---

> > ### Author Response · Authors · 2025-11-22
> >
> > https://iclr.cc/Conferences/2026/CallForPapers
> >
> > states "applications to neuroscience & cognitive science" as one of
> > the Subject Areas.
> >
> > A search for "EEG" on openreview.net returns 75 papers submitted to
> > ICLR 2026.
> >
> > The reason that we included quotes from Palazzo et al. (2024) along with adjacent refuting quotes
> > from Li et al. (2021), Ahmed et al. (2021, 2022), and
> > Bharadwaj et al. (2023) is that most of the refutations do not require
> > familiarity with the work. Many of the false statements in Palazzo et
> > al. (2024) are self-evident from the plain English. Consider, for
> > example, Section 2 lines 026-053. Palazzo et al. (2024) discuss a phenomenon
> > that occurs at the 300-400 ms level. But the trials in Ahmed et
> > al. (2021) last 3000 ms. Similarly, Section 6, line 127-165 provides a
> > quote where Palazzo et al. (2024) claim that Bharadwaj et al. (2023)
> > only presented results for a single subject from a single dataset when
> > the quote from Bharadwaj et al. (2023) shows that this was done for a
> > second dataset with six subjects.
> >
> > A few of the issues require minor attention to technical results. The
> > quotes from Palazzo et al. (2024) on lines 172-182 and 187-189
> > claim that the supertrial method of Bharadwaj et al (2023) attenuates
> > higher-frequency bands. But Fig. 1 shows that the raw trials are the
> > lowest curve while increasing the number of trials included in
> > supertrials successively amplifies the signal, preserving the spectral
> > characteristics. This is clear by simple inspection.
> >
> > Further, Table 1 shows that with supertrials, LSTM and EEGChannelNet,
> > the methods from Spampinato et al. (2017) and Palazzo et al. (2021),
> > are at chance for all numbers of trials per supertrial, but SVM, 1D
> > CNN, EEGNet, and SyncNet, methods evaluated in Bharadwaj et
> > al. (2023), are all above chance for many numbers of trials per
> > supertrial. This is also clear by simple inspection.
> >
> > Ww understand and appreciate that it takes both time and familiarity
> > with the literature to review submissions. We respectfully ask the AC
> > to find people with that time and familiarity to review our
> > submission. The stakes are high given the amount of flawed work that
> > is being submitted and published, even to ICLR.

---

### Author Response · Authors · 2025-11-21
**Specific relevance and significance to ICLR and the ML community**

It is important, if not imperative, for the community to publish this
work. Without it, the community continues to submit and publish more
flawed work at a growing rate. Fifty new papers papers have been
published since the flaw was first reported in prominent venues: once
in CVPR (Ahmed et al. 2021) and three times in TPAMI (Li at al. 2021,
Ahmed et al. 2022, Bharadwaj et al 2023).

Some recent flawed work has been published even by the ML community in
top ML venues, despite awareness of the issue: Liu et al. (2024) in
NeurIPS collects a new dataset that suffers from the block
confound. While the authors cite Li et al. (2021) and Ahmed et al
(2021), they fail to appreciate (or maybe hide the fact) that their
work is confounded.

Some recent flawed work has even been submitted to ICLR 2025 (and
apparently resubmitted to ICLR 2026 despite reviewer warnings). It
appears that even the reviewer pool of ICLR is unaware of the severity
of the confound.

https://openreview.net/forum?id=ejVuTFFkl6&noteId=zafmRtlFw1

collects a new dataset that suffers from the block confound. While the
authors again cite Li et al. (2021), they incorrectly claim that their
dataset does not suffer from the confound. All four of the reviewers
point this out. While this submission was rejected, three of the
reviewers rated it as "Soundness: 3: good" and two of the reviewers
rated it as "Contribution: 3: good".

The apparent resubmission (18265) to ICLR 2026 again cites Li et
al. (2021) and again incorrectly claims that their dataset does not
suffer from the confound. Again, three of the four reviewers point out
that this work suffers from the block confound. And again, two of the
reviewers rate this as "Soundness: 3: good", one of the reviewers
rates this as "Contribution: 3: good", and one even rates this as
"Contribution: 4: excellent" and recommends acceptance.

We have a simple question for the reviewers, area chairs, and program
chairs: If one cannot publish refutations like this in ICLR, how else
do you propose we address the fact that there is a large and growing
body of flawed work being published?

---

### Author Response · Authors · 2025-11-21
**General response**

Simply stated, Palazzo et al. (2024) made numerous false claims about
Li et al. (2021), Ahmed et al. (2021, 2022), and Bharadwaj et
al. (2023) and the authors of Li et al. (2021), Ahmed et al. (2021,
2022), and Bharadwaj et al. (2023) have not been afforded the
opportunity to respond to those false claims in TPAMI.

We decided to submit to ICLR because we though that ICLR would be
willing to publish a response like this due to the open and robust
nature of the dialog between authors and reviewers in the author
response period.

We have a simple question for the reviewers, area chairs, and program
chairs: If neither TPAMI nor ICLR are unwilling to publish a
manuscript like this, where can this be published?

We agree that in an ideal world, this would be published in
TPAMI. However, TPAMI policy requires this to be a Comment, not a
regular paper. And IEEE Computer Society policy stipulates that
Comments are not peer reviewed. They are sent to the authors of the
commented on paper who have outsized influence as to whether the
Comment gets published. This introduces a conflict of interest that
has precluded publication of this work in TPAMI.

This is unfortunate. The history of science is replete with open
published debate that has occurred in many different fields
(e.g. Heliocentrism-Geocentrism, Leibniz-Clarke, Priestley-Lavoisier,
Wilberforce-Huxley, Boltzmann-Ostwald, Cuvier–Geoffroy, Kelvin-Stokes,
Bohr-Einstein, Shapley-Curtis, Hoyle Gamow, Chandrasekhar-Eddington,
Kosslyn-Pylyshyn, Drexler-Smalley, ...) As such debates often lead to
major paradigm shifts and repressing the debate (Galileo-Catholic
Church) can impede scientific progress, we suggest that ICLR should
seek to advance science by correcting flaws in the literature, not
impede science by allowing those flaws to flourish.

---

### Meta-Review · Area_Chair_ZSzB · 2026-01-08

**Summary:**

All reviewers concerned about the scope of this work and the page limit.

**Reviewer Concerns:**

All reviewers concerned about the scope of this work and the page limit.

**Reviewer Scores:**

No reviewer may increase the score.

---

### Decision · Program_Chairs · 2026-01-26

Reject